# Rapid Identification of Constituents in *Polygonatum cyrtonema* Hua Using UHPLC-Q-Exactive Orbitrap Mass Spectrometry

**DOI:** 10.3390/molecules30030723

**Published:** 2025-02-05

**Authors:** Qingrui Yang, Jieyao Ma, Shenlong Yan, Suyu Yang, Lingxuan Fan, Yanghui Huo, Bowen Gao, Wei Cai

**Affiliations:** 1School of Pharmacy, Baotou Medical College, Baotou 014000, China; qingruiyangjj@163.com (Q.Y.); 17332325609@163.com (S.Y.); flxuan930@163.com (L.F.); hyanghui818@163.com (Y.H.); 2School of Pharmaceutical Sciences, Hunan University of Medicine, Huaihua 418000, China; majieyao@hnmu.edu.cn (J.M.); 18143386418@163.com (S.Y.)

**Keywords:** *Polygonatum cyrtonema*, ultra-high-performance liquid chromatography-Q-exactive orbitrap mass spectrometry, chemical composition

## Abstract

*Polygonatum cyrtonema* Hua (PCH) belongs to the genus Polygonatum Mill of the Liliaceae family. As a traditional tonic herb, the rhizome of PCH has been widely used as a functional food and traditional Chinese medicine, mainly for the treatment of spleen and lung Qi deficiency, essence and blood deficiency, internal heat, and thirst. To further elucidate the unknown chemical composition of PCH, this study presents an analytical strategy using macroporous resin (D101) column chromatography combined with ultra-high-performance liquid chromatography-Q-Exactive Orbitrap mass spectrometry (UHPLC-Q-Exactive Orbitrap MS) for the characterization of PCH’s chemical composition. The PCH extracts were separated via D101 resin column chromatography in conjunction with reverse phase liquid chromatography (C18 column). They were then analyzed by Q-Exactive Orbitrap mass spectrometry utilizing parallel reaction monitoring (PRM) mode, diagnostic fragment ions (DFIs), and neutral loss (NL). A total of 153 compounds were identified through comparing the mass spectrometry data with standard references, the published literature, and public databases, including 40 alkaloids, 43 organic acids, 30 flavonoids, 17 saponins, and 23 other compounds; The result expands PCH’s chemical composition, enhancing our understanding of its therapeutic effects and quality assurance. At the same time, the strategy has the potential to show a wide range of applications in the chemical characterization of different samples.

## 1. Introduction

*Polygonatum cyrtonema* Hua (PCH), a genus Polyflorum of Liliaceae, was mainly distributed in Hunan, Guizhou, Jiangxi, and other vast southern regions [1]. PCH has been commonly utilized in functional foods and traditional Chinese medicine due to its properties of tonifying the kidney and spleen, invigorating qi, nourishing yin, and supporting lung function and enhancing body fluids. Modern pharmacological research indicates that PCH also exhibits effects, such as reducing blood glucose levels, demonstrating anti-tumor activity, and combating vascular dementia [2]. Currently, the investigation of PCH constituents primarily employs traditional separation and purification methods for identification. The disadvantage of these methods is that they are operationally complex and time-consuming. The major identified compounds to date include flavonoids, polysaccharides, and saponins, while reports on other components remain relatively limited [3]. Huang et al. identified 64 compounds from Polygonatum sibiricum Flower using HPLC-QTOF-MS/MS [4]. Thus, the study of the chemical components in PCH remains inadequate. Consequently, there is a requirement to establish a comprehensive approach for the swift analysis and recognition of PCH’s components. This would facilitate a better understanding of its material basis and enhance quality control measures.

Ultra-high-performance liquid chromatography-Q-Exactive Orbitrap/mass spectrometry (UHPLC-Q-Exactive Orbitrap MS) has been shown to be a very effective analytical tool for the rapid determination of chemical composition in plant extracts, showing a wide analytical range and powerful separation capabilities, and producing reliable qualitative results [5,6]. It also has the advantage of high sensitivity and low sample requirements, especially in cases where sample volumes are small or no reference compounds are available [7,8].

Macroporous resins (MPRs) are porous materials derived from organic polymers that selectively adsorb compounds based on their polarity, size, and shape. By adjusting solvent polarity and pH, the adsorption and desorption of target compounds can be optimized, enabling the effective separation of specific components from complex natural product mixtures. MPRs offer advantages such as simplicity, high capacity, low cost, high selectivity, and easy regeneration. Recent years have witnessed a rise in the use of MPRs for the purification of bioactive compounds from medicinal plants, such as phenolics, saponins, alkaloids, and flavonoids [9,10]. D-101 macroporous resin is a non-polar adsorbent based on a styrene-divinylbenzene copolymer, exhibiting superior separation efficiency for a wide range of bioactive compounds, including saponins, flavonoids, alkaloids, and phenolic acids.

In this study, we developed a systematic approach utilizing macroporous resin (D101) column chromatography combined with UHPLC-Q-Exactive Orbitrap for the identification of the chemical constituents in PCH. As a result, a total of 153 compounds, including 40 alkaloids, 43 organic acids, 30 flavonoids, 17 saponins, and 23 others. These findings facilitate a deeper understanding of the pharmacological effects and mechanisms of action of PCH, providing a crucial foundation for its future clinical application, quality control, and potential therapeutic uses.

## 2. Results

### 2.1. Establishment of Analytical Strategy

In order to systematically screen and identify the various chemical components of PCH, a strategy based on UHPLC-Q-exactive orbitrap MS combined with parallel reaction monitoring (PRM), diagnostic fragment ions (DFIs), and neutral loss (NL) was established. First, seven samples were obtained by heated reflux extraction combined with macroporous resin. The sample was injected into a UHPLC-Q-Exactive Orbitrap Mass Spectrometer, which provided high-resolution MS data. This instrument was also used to collect MS/MS (MS^2^) data for the trace components present in the sample through parallel reaction monitoring (PRM) scanning. The identification of the chemicals was achieved by comparing the obtained spectral data with those of authentic standards, database information systems (DFIS), normalized retention indices (NL), and the existing literature.

### 2.2. Profiling of the Chemical Composition of PCH by LC-MS/MS

In total, 153 compounds were identified in PCH, including 40 alkaloids, 43 organic acids, 30 flavonoids, 17 saponins, and 23 others. Among them, 33 compounds were accurately identified by the reference standard. The chromatographic and mass data of those detected constituents are listed in Appendix A and the high-resolution extracted ion chromatograms (HREICs) are shown in Appendix A.

#### 2.2.1. Characterization of the Alkaloids in PCH

Compounds **27**, **87**, and **93** were observed at 4.80, 7.14, and 7.20 min, identified as Sinomenine, N-nornuciferine, and roemerine, respectively, by comparing the retention time and MS data with reference standards. Compound **62**, which possessed the same quasi-molecular ions and characteristic fragment ions as N-nornuciferine, was characterized as being N-nornuciferine isomers [11].

Compounds **30** and **37** were eluted at 5.17 and 5.35 min, respectively, and possessed the same quasi-molecular ion [M − H]^−^ at *m*/*z* 215.0826. The main product ions at *m*/*z* 171.0917 were attributed to the loss of CO_2_ (44 Da), and the obtained product ions at *m*/*z* 142.0650 were attributed to the loss of C_2_H_3_O_2_N (73 Da). Therefore, they were identified as being 2,3,4,6-tetrahydro-1H-β-carboline-3-carboxylic acid isomer [12].

Compound **36** was eluted at 5.32 min, possessing the quasi-molecular ions [M + H]^+^ at *m*/*z* 490.2072 and the daughter ions at m/z 328.1536 obtained by the loss of a glucose residue (162 Da). Therefore, it was identified as the 11-glc-norisocorydine [13].

Compound **38** was eluted at 5.41 min, possessing the quasi-molecular ions [M − H]^−^ at *m*/*z* 285.1245 and the fragmentation at *m*/*z* 207.0920, generated by the sequential loss of the C_2_H_4_O_2_ group (60 Da) and H_2_O (18 Da). The daughter ion at *m*/*z* 195.0553 and 308.0915 was due to the loss of C_2_H_6_O_2_ (62 Da) and C_2_H_4_ (28 Da) in MS^2^. Therefore, it was identified as the 5(9H-β-carbolin-1-yl)-pentane-1,2,5-triol. Likewise, Compounds **54** and **69** were characterized as being coumaroyloctopamine isomers. Furthermore, Compounds **112**, **116**, and **148** were tentatively identified as the isomers of N-trans-feruloyltyramine isomer [12,14].

Compounds **39**, **42**, **44**, and **46** were eluted at 5.50, 5.67, 5.95, and 6.05 min, respectively, and possessed the same quasi-molecular ion [M]^+^ at *m*/*z* 328.1549. They were deduced as being a boldine isomer, according to the MS and MS/MS spectra [15].

Compound **40** was eluted at 5.50 min and possessed the quasi-molecular ion [M]^+^ at *m*/*z* 344.1862. The main daughter ion at *m*/*z* 299.1275 was attributed to the loss of a C_2_H_7_N radical (45 Da). The fragment ions were obtained at *m*/*z* 175.0751 due to the loss of a C_9_H_15_O_2_N radical (169 Da). Therefore, it was identified as being the zizyphusine + 2H [13].

Compound **45** had the quasi-molecular ions [M]^+^ at *m*/*z* 358.2018 and yielded the fragment ions at *m*/*z* 313.1413 due to the neutral loss of a C_2_H_7_N (45 Da). The product ions at *m*/*z* 192.1015 were obtained by the loss of C_10_H_14_O_2_ (166 Da) moieties from the main ion at *m*/*z* 307.0837. Therefore, it was identified as being the pareirarinea [13].

Compounds **47**, **56**, and **82** were eluted at 6.08, 6.55, and 7.03 min, respectively, and possessed the same quasi-molecular ion [M]^+^ at *m*/*z* 374.1604. The main daughter ions at *m*/*z* 329.1013 and 297.1104 were due to the loss of a C_2_H_7_N radical (45 Da) and CH_9_N_4_ radical (77 Da) in MS^2^. Therefore, they were identified as being the di-hydroxylation of the magnoflorine isomer [13].

Compound **48** was eluted at 6.10 min, possessing the quasi-molecular ions [M]^+^ at *m*/*z* 342.1705. The main product ions at *m*/*z* 192.1014 were attributed to the loss of a C_9_H_10_O_2_ radical (150 Da), and the obtained product ions at *m*/*z* 177.0783 were attributed to the loss of C_10_H_13_O_2_ (165 Da). Therefore, it was identified as the phellodendrine [16].

Compounds **49** and **66** were eluted at 6.25 and 6.72 min, respectively, and possessed the same quasi-molecular ion [M]^+^ at *m*/*z* 356.1862. The main daughter ions at *m*/*z* 311.1271 and 296.1038 were due to the loss of a C_2_H_7_N radical (45 Da) and C_3_H_10_N radical (60 Da) in MS^2^. Therefore, they were identified as being the menisperine isomer [16].

Compound **50** was eluted at 6.36 min, with the quasi-molecular ions [M + H] ^+^ at *m*/*z* 200.0706 and the fragment ion [M + H]^+^ at *m*/*z* 182.0597, indicating the neutral loss of H_2_O (18 Da). Therefore, according to the mzVault database, it was identified as the dictamnine isomer.

Compounds **60** and **94** were eluted at 6.58 and 7.24 min, respectively, and possessed the same quasi-molecular ion [M]^+^ at *m*/*z* 296.1651. The main daughter ion at *m*/*z* 251.1060 was attributed to the loss of a C_2_H_7_N radical (45 Da). The fragment ions were obtained at *m*/*z* 219.0799 due to the loss of a C_3_H_11_ON radical (77 Da), and they were identified as being C_1_-demethoxy-C_2_-dehydrox of magnoflorineisomer [13].

Compounds **65** and **81** were eluted at 6.72 and 7.00 min, respectively, and possessed the same quasi-molecular ion [M − H]^−^ at *m*/*z* 328.1190. The fragment ions at *m*/*z* 310.1085 and 175.0390 were due to the loss of a H_2_O (18 Da) and C_8_H_11_NO_2_ (153 Da) in MS^2^. The product ions at *m*/*z* 160.0395 were obtained by the loss of CH_3_ (18 Da) moieties from the fragment ion at *m*/*z* 175.0390. Therefore, they were identified as being N-feruloyloctopamine isomers [17].

Compound **70** was eluted at 6.75 min and possessed the quasi-molecular ion [M + H]^+^ at *m*/*z* 370.1649 and fragment ions at *m*/*z* 293.0803, 265.0858, 137.0591, and 58.0658. Therefore, according to the mzVault database, it was identified as being the allocryptopin.

Compounds **72** and **90** were eluted at 6.78 and 7.19 min, respectively, and possessed the same quasi-molecular ion [M + H]^+^ at *m*/*z* 294.1489. The main product ions at *m*/*z* 279.1247 were attributed to the loss of a CH_3_ radical (15 Da), and the obtained product ions at *m*/*z* 250.0938 were attributed to the loss of CO_2_ (44 Da). Therefore, they were identified as being dehydronuciferine isomers [18].

Compound **83** was eluted at 7.05 min, possessing quasi-molecular ions [M]^+^ at *m*/*z* 356.1498 and characteristic fragment ions at 311.1269, 279.1006, 251.1061, and 58.0658. Therefore, it was identified as the C_5_-methylene to the ketone of magnoflorine [13].

Compound **85** was eluted at 7.08 min, possessing quasi-molecular ions [M]^+^ at *m*/*z* 344.1498 and characteristic fragment ions at 283.0959, 265.0852, and 58.0659. Therefore, it was identified as the N-CH_3_-hydroxylation of the C_2_-O-demethylation of magnoflorine [13].

Compound **91** was eluted at 7.19 min, possessing quasi-molecular ions [M + H]^+^ at *m*/*z* 308.1281 and yielding fragment ions at *m*/*z* 249.0903 due to the neutral loss of CH_3_ON_2_(15 Da). The daughter ion at *m*/*z* 219.0809 was due to the loss of C_2_HO_3_N (87 Da) in MS^2^. Therefore, it was identified as N-acetylanonaine [13].

Compound **92** was eluted at 7.19 min, possessing quasi-molecular ions [M]^+^ at *m*/*z* 338.1392. Therefore, it was identified as Columbamine. The main daughter ions at *m*/*z* 322.1071 and 323.1144 were due to the loss of an OH radical (16 Da) and CH_3_ radical (15 Da) in MS^2^ [19].

Compound **102** was eluted at 7.57 min and possessed the quasi-molecular ion [M + H]^+^ at *m*/*z* 623.3116 and fragment ions at *m*/*z* 251.1058 and 250.0982. Therefore, according to the mzVault database, it was identified as being tetrandrine [20].

Compound **145** was eluted at 14.62 min, possessing quasi-molecular ions [M]^+^ at *m*/*z* 322.1079, and yielded fragment ions at *m*/*z* 307.0829 due to the neutral loss of a CH_3_ (15 Da). The product ions at *m*/*z* 279.0887 were obtained by the loss of CO (28 Da) moieties from the main ion at *m*/*z* 307.0837. Therefore, it was identified as groenlandicine [13].

Compound **147** was found at 14.70 min, possessing quasi-molecular ions [M]^+^ at *m*/*z* 324.1236, and was tentatively identified as demethyleneberberine [19]. The main product ions at *m*/*z* 309.0978 were attributed to the loss of a CH_3_ radical (15 Da), and the obtained product ions at *m*/*z* 280.0971 were attributed to the loss of CO_2_ (44 Da) from the product ions at *m*/*z* 324.1236 [13].

#### 2.2.2. Characterization of the Organic Acids in PCH

Compounds **10**, **24**, **25**, **26**, **34**, **35**, **41**, **55**, **58**, **75**, **86**, **101**, **108**, and **120** were observed at 0.92, 3.73, 4.05, 4.25, 5.24, 5.32, 5.65, 6.54, 6.57, 6.91, 7.11, 7.54, 7.94, and 8.79 min and were identified as L-(-)-malic acid, 3,4-dihydroxybenzoic acid, L-tryptophan, neochlorogenic acid, chlorogenic acid, cryptochlorogenic acid, caffeic acid, p-Coumaric acid, octanedioic acid, ferulic acid, isoferulic acid, azelaic acid, salicylic acid, and abscisic acid, respectively, by comparing the retention time and MS data with reference standards.

Compounds **32** and **121** possessed the same quasi-molecular ion and characteristic fragment ions as L-tryptophan. Therefore, they were characterized as being L-tryptophan isomers. Likewise, Compounds **28** and **29** were deemed to be the isomers of salicylic acid, Compounds **52** and **126** were deemed to be the isomers of ferulic acid, Compounds **59**, **104**, and **125** were deemed to be the isomers of caffeic acid, and Compounds **71** and **105** were deemed to be the isomers of p-Coumaric acid. Furthermore, Compound **113** was deemed to be the isomer of abscisic acid.

Compound **1** was eluted at 0.68 min, possessing the quasi-molecular ions [M + H]^+^ at *m*/*z* 175.1190. Therefore, it was identified as arginine acid. The main daughter ions at *m*/*z* 158.0922 and 130.0974 were due to the loss of an H_3_N radical (17 Da) and CH_3_ON radical (45 Da) in MS^2^, respectively [21].

Compound **4** was detected at 0.83 min with the same empirical molecular formula, C_5_H_9_NO_2_, matched to that of proline. The main daughter ions at *m*/*z* 98.0604 and 88.0396 were due to loss of a H_2_O radical (18 Da) and 2CH_2_ radical (28 Da) in MS^2^ [22].

Compounds **6** and **14** were eluted at 0.91 and 1.30 min, possessing the quasi-molecular ions [M − H]^−^ at *m*/*z* 191.0197 and fragment ions at 111.0074, 87.0074, and 85.0280. According to the references [23], they were identified as the citric acid isomer.

Compounds **7** and **11** were eluted at 0.91 and 1.27 min, possessing the quasi-molecular ions [M + H]^+^ at *m*/*z* 130.0863, matched to that of pipercoic acid isomers. The dominant product ions at *m*/*z* 84.0812 and 56.0501 were attributed to a neutral loss of CH_2_O_2_ (46 Da) and C_3_H_6_O_2_ (74 Da), respectively.

Compounds **8**, **13**, and **22** at *m*/*z* 166.0863 had the same molecular formula C_9_H_11_NO_2_ and appeared at retention times (tR) of 0.91, 1.29, and 2.53 min. Based on these MS^2^ data, Compounds **8**, **13**, and **22** were identified as phenylalanine isomers and consistent with reference [21].

Compounds **9** and **16** were eluted at 0.91 and 1.31 min, possessing the quasi-molecular ions [M − H]^−^ at *m*/*z* 180.0666 and the main daughter ion at *m*/*z* 163.0390, which were due to the loss of a OH radical (17 Da). Therefore, according to the mzVault database, they were identified as the L-tyrosine isomer.

Compound **15** was eluted at 1.30 min, possessing the quasi-molecular ions [M + H] ^+^ at *m*/*z* 130.0499. The main daughter ions at *m*/*z* 84.0812 and 70.0657 were due to the loss of a CH_2_O_2_ radical (46 Da) and CO_3_ radical (60 Da) in MS^2^. Therefore, it was identified as the L-pyroglutamic acid isomer [12].

Compound **31** was eluted at 5.17 min, with the quasi-molecular ion [M − H]^−^ at *m*/*z* 341.0878 (C_15_H_18_O_9_) and their fragment ions [M − H]^−^ at *m*/*z* 101.0237, 135.0449, 161.0597, and 179.0347, respectively, indicating that they possessed a caffeic acid moiety. Considering the neutral loss of hexoside (162 Da), they were identified as caffeic acid-hexoside [24].

Compound **133** had the quasi-molecular ions [M − H]^−^ at *m*/*z* 329.2333, and yielded fragment ions at *m*/*z* 293.2126 due to the neutral loss of 2H_2_O (36 Da). Therefore, it was identified as the tianshic acid isomer [9]. The product ions at *m*/*z* 211.1334 were obtained by the loss of C_6_H_14_O_2_ (118 Da) moieties from the parent ion.

Compounds **149**, **151**, and **152** were eluted at 15.04, 15.46, and 15.90 min, respectively, possessing the same quasi-molecular ions [M + H]^+^ at *m*/*z* 279.2318, and were tentatively identified as being α-linolenic acid isomers [25]. The main product ions at *m*/*z* 261.2212 and 243.2095 were attributed to the successive loss of a H_2_O radical (15 Da).

#### 2.2.3. Characterization of the Flavonoids in PCH

Compounds **53**, **57**, **64**, **67**, **74**, **78**, **84**, **89**, **96**, **100**, **119**, **122**, **123**, **127**, **128**, **135**, **139** and **146** were observed at 6.45, 6.56, 6.67, 6.72, 6.87, 6.94, 7.07, 7.17, 7.30, 7.49, 8.65, 8.90, 8.92, 10.15, 10.60, 12.34, 13.72, and 14.66 min and were identified as rutin, kaempferol 3-glucorhamnoside, hyperoside, luteolin-7-O-glucoside, nicotiflorin, taxifolin, astragalin, hesperidin, apigenin-7-glucoside, phlorizin, liquiritigenin, luteolin, quercetin, naringenin, kaempferol, formononetin, wogonin, and proanthocyanidin, respectively, by comparing the retention time and MS data with reference standards.

Compounds **124** and **134** possessed the same quasi-molecular ion and characteristic fragment ions as Wogonin. Therefore, they were characterized as being wogonin isomers.

Compound **51** was eluted at 6.44 min, possessing the quasi-molecular ions [M − H]^−^ at *m*/*z* 433.1140 and the typical daughter ions at *m*/*z* 271.061, obtained by the loss of a glucose residue (162 Da). However, the characteristic fragment ions at *m*/*z* 151.0028 and 119.0491 indicated that they possessed a naringenin group. Therefore, it was identified as the Naringenin 7-O-glucoside [10].

Compounds **63** and **76** were eluted at 6.66 and 6.92 min, possessing the quasi-molecular ions [M + H]^+^ at *m*/*z* 317.0655. The main daughter ion at *m*/*z* 302.0415 and 285.0389 were due to the loss of a CH_3_ radical (15 Da) and CH_4_O radical (32 Da) in MS^2^. Therefore, they were identified as the isorhamnetin isomer [26].

Compound **77** was eluted at 6.92 min, possessing the quasi-molecular ions [M − H]^−^ at *m*/*z* 623.1618 and the typical daughter ions at *m*/*z* 315.0509, obtained by the loss of one rutinose moiety (308 Da). The characteristic fragment ions at *m*/*z* 271.0257 and 151.0028 indicated that they possessed a quercetin group. Therefore, it was identified as the methyl-quercetin-O-rutinoside.

Compounds **88**, **95**, **99**, and **103** were eluted at 7.17, 7.30, 7.38, and 7.63 min, possessing the quasi-molecular ions [M + H]^+^ at *m*/*z* 303.0863 and fragment ions at 149.0596, 131.0489, and 123.0439. Therefore, according to the mzVault database, they were identified as the hematoxylin isomer.

Compound **140** had the quasi-molecular ions [M − H]^−^ at *m*/*z* 313.1081, and yielded fragment ions at *m*/*z* 295.2277 due to the neutral loss of a H_2_O (18 Da). The product ions at *m*/*z* 207.0655 were obtained by the loss of C_3_H_14_N_4_ (106 Da) moieties. Therefore, it was identified as 6, 8-dimethyl-4′,5, 7-trihydroxyisoflavanones [25,27].

Compound **141** was eluted at 14.10 min, possessing the quasi-molecular ions [M − H]^−^ at *m*/*z* 329.1031. The main daughter ions at *m*/*z* 311.2231 and 139.0389 were due to the loss of a CH_3_ radical (15 Da) and CH_4_O radical (32 Da) in MS^2^. Therefore, it was identified as the 4′,5,7-trihydroxy-8-methoxyhomoisoflavon [18].

#### 2.2.4. Characterization of the Aponins in PCH

Compounds **61** and **73** were eluted at 6.64 and 6.78 min, possessing the quasi-molecular ions [M − H]^−^ at *m*/*z* 1093.5072. The fragment ions at *m*/*z* 913.4439 and 769.4042 were generated by the sequential loss of a C_6_H_10_O_5_ (162 Da) in MS^2^. Therefore, they were identified as the kingianoside E isomer [12].

Compound **79** was eluted at 6.95 min, possessing the quasi-molecular ions [M − H]^−^ at *m*/*z* 931.4544. The fragment ions at *m*/*z* 769.4042 and 607.5002 were generated by the sequential loss of a C_6_H_10_O_5_ (162 Da) in MS^2^. Therefore, it was identified as the kingianoside C [18,28].

Compounds **97** and **107** were eluted at 7.37 and 7.92 min, possessing the quasi-molecular ions [M − H]^−^ at *m*/*z* 1237.5495. Due to the continuous separation of C_6_H_10_O_5_, the fragment ions at *m*/*z* 1075.4954, 913.4560, and 751.3934 were obtained. On this basis, the fragment ion at *m*/*z* 571.3272 was obtained by the further separation of a molecule of C_6_H_12_O_6_. Therefore, they were identified as the kingianoside Z isomer [29].

Compounds **98** and **106** were eluted at 7.37 and 7.84 min, possessing the quasi-molecular ions [M − H]^−^ at *m*/*z* 1241.5808. The fragment ions at *m*/*z* 1079.6985, 917.6189, and 755.5413 were generated by the sequential loss of a C_6_H_10_O_5_ (162 Da) in MS^2^. Therefore, they were identified as the flavopiridol glycosides isomer [25].

Compounds **109** and **114** were eluted at 7.94 and 8.36 min, possessing the quasi-molecular ions [M + H]^+^ at *m*/*z* 593.3684. The fragment ions at *m*/*z* 269.1902 were generated by the sequential loss of 2C_6_H_10_O_5_ (324 Da) in MS^2^. Therefore, they were identified as the huangjinoside D isomer [12,29,30].

Compound **110** was eluted at 8.12 min, possessing the quasi-molecular ions [M − H]^−^ at *m*/*z* 1075.4966 and the fragment ions at *m*/*z* 913.4478 due to the neutral loss of a C_6_H_10_O_5_ (162 Da). Therefore, it was identified as cyrtonemoside A [12].

Compound **117** had the quasi-molecular ions [M − H]^−^ at *m*/*z* 1031.5421 and yielded fragment ions at *m*/*z* 869.4947 due to the neutral loss of a C_6_H_10_O_5_ (162 Da). Therefore, according to the mzVault database, it was identified as pseudoprotodioscin.

Compounds **137** and **142** had the quasi-molecular ions [M + H]^+^ at *m*/*z* 429.2999 and yielded fragment ions at *m*/*z* 411.2886 due to the neutral loss of an H_2_O (18 Da). Therefore, they were identified as neoruscogenin isomers [31].

Compound **138** was eluted at 13.57 min, possessing the quasi-molecular ions [M − H]^−^ at *m*/*z* 913.4439 and fragment ions at *m*/*z* 411.2886 due to the neutral loss of a C_6_H_10_O_5_ (162 Da). The fragment ions at *m*/*z* 383.1182 and 221.0659 were obtained from the internal fracture of the parent ion sugar chain. Therefore, it was identified as the pratioside D1 [12].

Compound **143** was eluted at 14.22 min, possessing the quasi-molecular ions [M − H]^−^ at *m*/*z* 751.3910 and fragment ions at 161.0446, 179.0553, 131.0337, 101.0231, and 113.0231. Therefore, it was identified as the kingianoside A [29].

#### 2.2.5. Other Chemical Constituents in PCH

Compounds **43** and **153** were observed at 5.83 and 22.63 min and were identified as riboflavin and betulinic acid, respectively, by comparing the retention time and MS data with reference standards.

Compound **2** was eluted at 0.81 min, possessing the quasi-molecular ions [M + H]^+^ at *m*/*z* 112.0505. Therefore, it was identified as cytosine [32]. The main daughter ion at *m*/*z* 95.0242 was attributed to the loss of an OH (17 Da), which further gave rise to the product ions at *m*/*z* 94.0402 due to the loss of an H_2_O radical (15 Da).

Compounds **5** and **17** were eluted at 0.90 and 1.36 min, respectively, with the quasimolecular ion [M + H]^+^ at *m*/*z* 268.1040 and the fragmentation ion [M + H]^+^ at *m*/*z* 136.0615, resulting from the neutral loss of 132 Da and were identified as the loss of C_5_H_8_O_4_. Therefore, they were identified as the adenosine isomer [29].

Compounds **18**, **20**, **23**, and **33** were eluted at 1.43, 1.54, 3.51, and 5.22 min, respectively, and possessed the same quasi-molecular ions [M + H]^+^ at *m*/*z* 248.1492 and the main fragment ion at *m*/*z* 230.1383 and 212.1284, generated by the sequential loss of the H_2_O group (18 Da). According to the mzVault database, they were identified as the pantothenic acid isomer.

Compound **19** was eluted at 1.48 min, possessing the quasi-molecular ions [M + H]^+^ at *m*/*z* 284.0989 and the fragment ions at *m*/*z* 152.0564 due to the neutral loss of a C_9_H_8_O (132 Da). Therefore, according to the mzVault database, it was identified as isoguanosine.

Compounds **68**, **80**, **111**, and **115** were eluted at 6.73, 7.00, 8.13, and 8.49 min, respectively, possessing the quasi-molecular ions [M + H]^+^ at *m*/*z* 177.0546. The main daughter ions at *m*/*z* 149.0594 and 145.0281 were due to loss of a CO radical (28 Da) and CH_4_O radical (32 Da) in MS^2^. Therefore, they were identified as the 4-methylumbelliferone isomer [33].

Compounds **129**, **130**, and **131** had the quasi-molecular ions [M + H]^+^ at *m*/*z* 235.1693 and yielded fragment ions at *m*/*z* 199.1479 due to the neutral loss of 2H_2_O (36 Da). The product ions at *m*/*z* 189.1633 and 157.1009 were obtained by the loss of CH_2_O_2_ (46 Da) and C_3_H_10_O_2_ (78 Da) moieties. Hence, according to the mzVault database, they were deduced as being the curcumenol isomer.

Compounds **118** and **136** were eluted at 8.55 and 13.31 min, respectively, possessing the same quasi-molecular ions [M − H]^−^ at *m*/*z* 299.0924, and were deduced as being 5,7-dihydroxy-8-methyl-3-(4-hydroxybenzyl)-chromate-4-one isomers. The main daughter ion at *m*/*z* 281.1767 and 193.0502 was due to loss of a H_2_O radical (18 Da) and C_7_H_6_O radical (106 Da) in MS^2^, respectively [25].

Compound **132** was eluted at 11.54 min and yielded a deprotonated ion [M − H]^−^ *m*/*z* 315.0874 and fragment ions at *m*/*z* 205.0500, 193.0497, and 139.0389. Based on these MS data, compound **132** was identified as the odoratumone B isomer and was consistent with reference [18].

## 3. Discussion

Huang et al. identified 64 compounds from Polygonatum sibiricum Flower using HPLC-QTOF-MS/MS [4]. In our study, mass spectrometry analysis of the aqueous extract of Polygonatum identified 80 compounds. After pre-treatment with D101 macroporous resin followed by mass spectrometry, all 80 original components were detected, and an additional 73 compounds were identified. D101 macroporous resin is particularly suitable for analyzing compounds such as saponins, flavonoids, alkaloids, and phenolic acids [34]. It selectively adsorbs target compounds while reducing impurities, thereby simplifying the sample matrix and minimizing background interference. This enhances the sensitivity and accuracy of subsequent analyses. The aqueous extract was fractionated into six extracts (W, T, S, F, E, N) through selective elution using different ethanol concentrations. This process effectively enriches the compounds of interest, reduces background noise, and enables the more precise detection of trace or low-abundance components. The use of macroporous resin chromatography coupled with UHPLC-Q-Exactive Orbitrap MS allowed for the identification of more compounds from PCH, leading to a more efficient method for characterization of the chemical composition in PCH. The macroporous resin separation experiment takes approximately 8 h, and the drying time is also about 8 h. Including the time for UHPLC-Q-Exactive Orbitrap MS analysis, the entire experiment can be completed within 24 h. This method is simple to operate and efficient.

Among the six extracts (W, T, S, F, E, and N), W, T, and S mainly contain compounds with higher polarity, such as organic acids. F primarily consists of organic acids and alkaloids. E is predominantly composed of alkaloids and saponins, which have relatively lower polarity compared to organic acids. The flavonoids, characterized by relatively lower polarity, are mainly concentrated in N.

A total of 153 compounds were identified through comparing the mass spectrometry data with standard references, the published literature, and public databases, including 40 alkaloids, 43 organic acids, 30 flavonoids, 17 saponins, and 23 other compounds. Of these, 134 were detected for the first time in PCH, significantly enriching our understanding of its chemical composition.

The compounds identified in this study include flavonoids, alkaloids, organic acids, saponins, and other compounds. The basic chemical structure of flavonoids consists of two benzene rings connected by a three-carbon bridge, forming a C6-C3-C6 structure. Depending on the oxidation state of the central three-carbon bridge and the position and type of substituents, flavonoids can be classified into several subclasses, such as flavones, flavonols, flavanones, isoflavones, and anthocyanins. Flavonoids are widely found in various plants and have a wide range of pharmacological activities, including anti-inflammatory, antioxidant, hypoglycemic, and neuroprotective effects [35,36]. For example, rutin has pharmacological properties, such as antioxidant, anti-apoptotic, and anti-inflammatory properties, as well as nephroprotective and blood vasculature properties [37,38]. The analogue of rutin, dihydroquercetin, shows similar activity to rutin, with a more potent antioxidant effect [39]. Kaempferol and some glycosylated derivatives have important neuroprotective effects, mainly through the inhibition of pro-inflammatory cytotoxicity and the activity of important inflammatory pathways, such as NF-kB, p38MAPK, and AKT, which results in a comprehensive anti-inflammatory effect that ultimately contributes to the protective effects on the brain [40].

Organic acids are compounds that contain one or more carboxyl groups (-COOH) and can release protons (H⁺), thereby exhibiting acidic properties. Common examples include acetic acid, lactic acid, citric acid, malic acid, and oxalic acid. Phenolic acids are one of the most common polyphenols, with cardiac protective, neuroprotective, anti-cancer, anti-diabetic, and anti-allergic properties [41]. Ferulic acid is a hydroxyl derivative derived from plants and has a wide range of pharmacological activities, including immune modulation, tumor and diabetes prevention, and anti-hypertension [42]. Chlorogenic acid, cryptochlorogenic acid, and neochlorogenic acid have been reported to have good anti-inflammatory, antioxidant, anticancer, and antimicrobial properties [43,44,45,46].

Alkaloids contain at least one nitrogen atom, typically in the form of an amine, which gives most alkaloids their basic (alkaline) nature. Alkaloids are the active components of many traditional Chinese medicines and have a wide range of biological properties. For example, sinomenine is one of the most important pharmacologically active compounds among the isoquinoline alkaloids, with analgesic, antiarrhythmic, and anti-inflammatory properties and neuroprotective potential in hippocampal cells [47,48]. Roemerine is a kind of aporphine that has certain neuroprotective functions and antibacterial activity [49,50].

Saponins consist of two parts: a polar glycoside portion and a nonpolar aglycone. The aglycone is typically a triterpene or spirostane compound. Saponin is a class of glycoside, the aglycone of which are triterpene or spirosterane compounds, which are mainly distributed in higher plants on land and have significant pharmacological activities. Neoruscogenin, a steroid saponin, is osteoprotective in rats with ovariectomy-induced osteoporosis [51]. The study found that pseudoprotodioscin administration alleviated atherosclerotic lesions by lowering total cholesterol in ovariectomized apoE^−/−^ mice fed a high-cholesterol diet [52]. These compounds may be the active ingredients of PCH and help in the study of its pharmacological activity.

## 4. Materials and Methods

### 4.1. Chemicals and Reference Standards

Methanol and acetonitrile (HPLC-grade) were provided by Merck (Branchburg, NJ, USA). The LC–MS-grade formic acid was purchased from Thermo Fisher Scientific (Carlsbad, CA, USA). D-101 macroporous adsorption resin was purchased from Tianjin Haiguang Chemical Co., Ltd. (Tianjin, China). Water used as the mobile phase solvent was obtained from Watson Water (Guangzhou, China). Other solvents were of an analytical grade. Detailed information regarding the 37 standards is listed in Appendix A. The rhizome of PCH (20220607) was purchased from Hunan Bestcome Traditional Medicine Co., Ltd. (Huaihua, China).

### 4.2. Preparation of Standard and Sample Solutions

The dried PCH were powdered (50 g) and heated for reflux extraction at 80 °C for 2 h with 1000 mL of water. After three extractions, the extracts were combined and filtered; the filtrate was alcohol precipitated with 30%, 50%, 70%, and 90% ethanol, centrifuged, and the supernatant was evaporated under reduced pressure and lyophilized to give a light yellow sample named A. Then, a total of 2 g of sample A was separated by chromatography using 200 g D101 macroporous resin, eluting sequentially with two column volumes of water, 10%, 30%, 50%, 70%, and 90% ethanol, and the fractions were evaporated under reduced pressure and lyophilized to give samples named W, T, S, F, E, and N, respectively.

Each stage fraction (1 mg) was dissolved in 1 mL methanol-water (7:3, *v*/*v*) for 30 min under ultrasound. The obtained solution was centrifuged at 12,000 rpm for 20 min.

All reference standards were precisely weighed and then dissolved in methanol. The solutions were stored in a refrigerator at 4 °C until needed for further analysis.

### 4.3. Instruments and LC–MS/MS Conditions

Qualitative analyses were carried out using a Q-Exactive Focus Orbitrap Mass Spectrometer (Thermo Fisher Scientific, Carlsbad, CA, USA). The nitrogen gas used in the mass spectrometry was supplied by a GN-A nitrogen generator (Greenville Scientific LLC, Beijing, China). The UHPLC separation was performed using a Thermo Scientific Syncronis C18 column (100 × 2.1 mm, 1.7 µm). The mobile phase was composed of 0.1% formic acid in water (A) and acetonitrile (B). The column was maintained at 45 °C with a flow rate of 0.28 mL/min. The gradient program was as follows: 0 min, 5% B; 2 min, 10% B; 5 min, 30% B; 10 min, 40% B; 12 min, 55% B; 20 min, 80% B; 25 min, 95% B; 26 min, 5% B; and 30 min, 5% B; the total runtime was 30 min. MS analysis was performed in both positive and negative ionization modes using electrospray ionization (ESI) in the scan range of *m*/*z* 100–1500. The additional conditions for MS analysis were as follows: sheath gas, 30; auxiliary gas, 10; spray voltage, 3.0 kV for negative ESI and 3.5 kV for positive ESI; capillary temperature, 320 °C; and auxiliary gas heater temperature, 350 °C. The MS1 spectra were acquired in full MS mode at a resolution of 35,000, whereas MS2 spectra were obtained by ddMS2 or parallel reaction monitoring (PRM) mode triggered by inclusion ions at a resolution of 17,500. The normalized collision energy (NEC) was configured at 30%, and the automatic gain control (AGC) target was set to 5.0 × 10^5^.

## 5. Conclusions

In this study, an effective strategy was developed for the detection of chemical components in PCH by UHPLC Q-Exactive Orbitrap MS in full scan mode combined with PRM. A total of 153 compounds were identified from PCH based on chromatographic retention, MS, and MS^2^, including 40 alkaloids, 43 organic acids, 30 flavonoids, 17 saponins, and 23 other compounds, 98 of which were identified from PCH for the first time. The results of the study provide support for a more in-depth study of the pharmacological basis of PCH.

## Data Availability

The original contributions presented in this study are included in the article/Appendix A. Further inquiries can be directed to the corresponding authors.

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
