# Peer review of "Rapid Identification of Constituents in Polygonatum cyrtonema Hua Using UHPLC-Q-Exactive Orbitrap Mass Spectrometry"

_molecules, 2025, doi:10.3390/molecules30030723_

Round 1
Reviewer 1 Report
Comments and Suggestions for Authors
Introduction: The introduction of the article was adequate, but I would add a paragraph discussing the D101 resin column chromatography, explaining its working mechanism and potential applications. Furthermore, the species name author is graphed in italic in both abstract and introduction which is not correct by the taxonomy rules.
Results: The results section presents the findings in a very raw manner, merely describing what is already largely presented in Table 1. I recommend removing this table as it is not relevant to the development of the article, since the table included in the supplementary material is sufficient. It is preferable to refer to the molecules directly by their names in the text; in several instances, the same molecule is referred to in a sentence by both its name and number unnecessarily. Furthermore, Figure 1 is not used as a basis for discussing any of the obtained results, being cited only once in the text. Therefore, it is recommended to relocated it to the supplementary material. Figure 2 also does not appear relevant to the text, being merely cited without discussion, and the reason for choosing these three molecules to represent their fragmentation pathways is not justified, making it incoherent amidst a range of 153 compounds included in the article. It is noteworthy that the representations of the fragmentations in Figure 2 are all incorrect, thus its removal is recommended.
Discussion: The discussion was not relevant or coherent with the article's proposal; the text is better suited for the introduction. This section should discuss the elution order of the compounds, comparisons with other metabolomic studies and analyses of the same plant or closely related species, and the effectiveness of the analytical method employed. Crucially, it should address how favourable or unfavourable the use of the resin was on the overall procedure compared to more common methods of concentration or substance separation, considering that its usage was the main differential factor of the article. A correlation of the annotated compounds with their known metabolic pathways could also be included.
Methods: The supplier of the resin should be indicated, or the method of its preparation. The specific part of the plant used was not mentioned and should be included, as this is crucial information for the study. The models of all equipment used should also be included, not just the UHPLC-Q-Orbitrap Mass Spectrometer. The gases used in the UHPLC-Q-Orbitrap Mass Spectrometer should be specified.
Conclusion: The conclusion states: "98 of which were isolated from PCH for the first time." This is incorrect, as no molecule was isolated. Only a chromatographic analysis was performed, which allowed for the annotation of the presence of some compounds, some of which had standards for comparison. This indicates that most annotations are, at most, level 2. For further information, it is recommended to consult the literature on molecule annotation in metabolomics.
Author Response
Dear editor and reviewers,
On behalf of my co-authors, I am submitting the enclosed material “Rapid Identification of Constituents in Polygonatum cyrtonema Hua Using UHPLC-Q-Exactive Orbitrap Mass Spectrometry” for possible publication in Molecules.
We have reviewed the revision version of the manuscript and approve it for publication. To the best of our knowledge and belief, this manuscript has not been published nor is it being considered for publication elsewhere.
Your comments were highly insightful and enabled us to greatly improve the quality of the manuscript. In the following pages are our point-by-point responses to each of the comments. The revision has been highlighted in red font in the manuscript.
Comments 1: Introduction: The introduction of the article was adequate, but I would add a paragraph discussing the D101 resin column chromatography, explaining its working mechanism and potential applications. Furthermore, the species name author is graphed in italic in both abstract and introduction which is not correct by the taxonomy rules.
Response 1: Thank you for your valuable and thoughtful comments. The introduction section of the article has been expanded to include the principles, overview, and applications of macroporous resins. Additionally, species names author throughout the text have been changed to regular font, removing the use of italics .
Comments 2: Results: The results section presents the findings in a very raw manner, merely describing what is already largely presented in Table 1. I recommend removing this table as it is not relevant to the development of the article, since the table included in the supplementary material is sufficient. It is preferable to refer to the molecules directly by their names in the text; in several instances, the same molecule is referred to in a sentence by both its name and number unnecessarily. Furthermore, Figure 1 is not used as a basis for discussing any of the obtained results, being cited only once in the text. Therefore, it is recommended to relocated it to the supplementary material. Figure 2 also does not appear relevant to the text, being merely cited without discussion, and the reason for choosing these three molecules to represent their fragmentation pathways is not justified, making it incoherent amidst a range of 153 compounds included in the article. It is noteworthy that the representations of the fragmentations in Figure 2 are all incorrect, thus its removal is recommended.
Response 2: Thank you for your valuable and thoughtful comments. Table 2S contains all the information from Table 1, which we have deleted according the reviewer's suggestion . Additionally, we have moved Figure 1 to the supplementary materials (Figure 1S). Figure 2 illustrated the fragmentation patterns of several representative compounds, and we have removed it. In descriptions related to compound identification, we uniformly start by stating the compound number, followed by an explanation of the identification basis, and finally specify the compound name. Following your suggestions, we have also made some adjustments to the language used in the descriptions.
Comments 3: Discussion: The discussion was not relevant or coherent with the article's proposal; the text is better suited for the introduction. This section should discuss the elution order of the compounds, comparisons with other metabolomic studies and analyses of the same plant or closely related species, and the effectiveness of the analytical method employed. Crucially, it should address how favourable or unfavourable the use of the resin was on the overall procedure compared to more common methods of concentration or substance separation, considering that its usage was the main differential factor of the article. A correlation of the annotated compounds with their known metabolic pathways could also be included.
Response 3: Thank you for your valuable and thoughtful comments. In the discussion section, we have added a discussion on the impact of pre-treatment with macroporous resin on the test results, as well as the characteristics of compound categories eluted from six fractions of the macroporous resin.
D101 macroporous resin is particularly suitable for analyzing compounds such as saponins, flavonoids, alkaloids, and phenolic acids. It selectively adsorbs target compounds while reducing impurities, thereby simplifying the sample matrix and minimizing background interference. This enhances the sensitivity and accuracy of subsequent analyses. The aqueous extract was fractionated into six extracts (W, T, S, F, E, N) through selective elution using different ethanol concentrations. This process effectively enriches the compounds of interest, reduces background noise, and enables more precise detection of trace or low-abundance components.
Comments 4: Methods: The supplier of the resin should be indicated, or the method of its preparation. The specific part of the plant used was not mentioned and should be included, as this is crucial information for the study. The models of all equipment used should also be included, not just the UHPLC-Q-Orbitrap Mass Spectrometer. The gases used in the UHPLC-Q-Orbitrap Mass Spectrometer should be specified.
Response 4: Answer:Thank you for your valuable and thoughtful comments. Information regarding the macroporous resin supplier, the part of the Polygonatum herb utilized, the model of the mass spectrometer, and the carrier gas employed has been added to the Methods section of this paper.
Qualitative analyses were carried out using Q-Exactive Focus Orbitrap Mass Spectrometer (Thermo Fisher Scientific, Carlsbad, CA, USA). The nitrogen gas used in the mass spectrometry was supplied by a GN-A nitrogen generator (Greenville Scientific LLc, Beijing, China). The rhizome of PCH (20220607) was purchased from Hunan Bestcome Traditional Medicine Co., Ltd. (Hunan, China).
Comments 5: Conclusion: The conclusion states: "98 of which were isolated from PCH for the first time." This is incorrect, as no molecule was isolated. Only a chromatographic analysis was performed, which allowed for the annotation of the presence of some compounds, some of which had standards for comparison. This indicates that most annotations are, at most, level 2. For further information, it is recommended to consult the literature on molecule annotation in metabolomics.
Response 5: Thank you for your valuable and thoughtful comments.
Mass spectrometry primarily identifies compounds using their MS1 and MS2 information, rather than isolating them. In this sentence, we have replaced the word "isolated" with "identified".
The MSI classification and references have been added in Table 2S. The MSI level distinction is explained in turn below.
MSI 1. Identified compounds (A minimum of two independent and orthogonal data relative to an authentic compound analyzed under identical experimental conditions are proposed as necessary to validate non-novel metabolite identifications). The sample and the real standard compound were analyzed under the same liquid phase and mass spectrometry conditions, to identify the compound by comparing its retention time, MS1, MS2. For example, nicotiflorin (#74) be classified as MSI1.
MSI 2. Putatively annotated compounds (e.g. without chemical reference standards, based upon physicochemical properties and/or spectral similarity with public/commercial spectral libraries). The compounds were identified by comparing the MS1 and MS2 of the compounds with the data in the article and the database. For example, dehydronuciferine (#72) be classified as MSI2.
MSI 3. Putatively characterized compound classes (e.g. based upon characteristic physicochemical properties of a chemical class of compounds, or by spectral similarity to known compounds of a chemical class). It is not possible to identify its specific structure, but it can only be judged that there may be an isomer relationship with other compounds. For example, kingianoside E isomer (#73) be classified as MSI3.
MSI 4. Unknown compounds—although unidentified or unclassified these metabolites can still be differentiated and quantified based upon spectral data. The secondary mass spectrometry of the compounds and the comparison of the article and database data inferred the presence of partial structures of the compounds identified compounds.
Reviewer 2 Report
Comments and Suggestions for Authors
Dear Authors
The paper is well written. The only suggestion is to expand the discussion with a more detailed explanation of the properties of the identified compounds.
Author Response
Dear editor and reviewers,
On behalf of my co-authors, I am submitting the enclosed material “Rapid Identification of Constituents in Polygonatum cyrtonema Hua Using UHPLC-Q-Exactive Orbitrap Mass Spectrometry” for possible publication in Molecules.
We have reviewed the revision version of the manuscript and approve it for publication. To the best of our knowledge and belief, this manuscript has not been published nor is it being considered for publication elsewhere.
Your comments were highly insightful and enabled us to greatly improve the quality of the manuscript. In the following pages are our point-by-point responses to each of the comments. The revision has been highlighted in red font in the manuscript.
Comments 1: The paper is well written. The only suggestion is to expand the discussion with a more detailed explanation of the properties of the identified compounds.
Response 1: Thank you for your valuable and thoughtful comments. The characteristics of each class of compounds have been added to the discussion section of the article.
Flavonoids: The basic chemical structure of flavonoids consists of two benzene rings connected by a three-carbon bridge, forming a C6-C3-C6 structure. Depending on the oxidation state of the central three-carbon bridge and the position and type of substituents, flavonoids can be classified into several subclasses, such as flavones, flavonols, flavanones, isoflavones, and anthocyanins.
Organic acids: Organic acids are compounds that contain one or more carboxyl groups (-COOH) and can release protons (H⁺), thereby exhibiting acidic properties. Common examples include acetic acid, lactic acid, citric acid, malic acid, and oxalic acid.
Alkaloids: Alkaloids contain at least one nitrogen atom, typically in the form of an amine, which gives most alkaloids their basic (alkaline) nature.
Saponins: Saponins consist of two parts: a polar glycoside portion and a nonpolar aglycone. The aglycone is typically a triterpene or spirostane compound.
Reviewer 3 Report
Comments and Suggestions for Authors
In this manuscript, Yang and co-workers described the identification of 153 species contained in Plygonatum cyrtonema Hua (PCH) by the combination of chromatographic separations and Mass Spectrometry determination in order to unravel the basis of medical properties of this PCH. Despite the valuable work on species identification based on MS results, I still have some questions that need to be resolved.
1. The introduction section explains the main aim of the deeper understanding of the medical properties of PCH through the species identification contained in this herb. However, there is a lack of information on previous efforts made by other researchers to identify the composition of PCH. A brief review of previous experiments that were attempting to determine some of these compounds and the techniques used should be included to better understand the state of the art and the novelty of this work.
2. Figure 1 shows the six high-resolution extracted ion chromatograms of the six extracts obtained after separation on D101 resin. However, there is no indication of which extract (labelled W, T, S, F, E and N in section 4.2) corresponds to each of the chromatograms shown (A-F).
3. Furthermore, the description of each chromatogram (A-F) used in the figure caption includes the numbers of the masses detected, but it is somewhat confusing to use this as a label. In addition, there are peaks whose m/z are given in Table 1 but are not mentioned in these labels in Figure 1.
4. Also in Figure 1, the axis labels and values are not visible because they are too small. This makes identification difficult.
5. The methodological section explained the sample preparation procedure. However, despite the explanation of the two separation techniques used (D101 macroporous resin and reversed phase chromatography), the total time required to carry out the whole chromatographic analyses, 6 extract D101 isolation and sequential UHPLC-Q-Orbitrap MS, was not given.
Author Response
Dear editor and reviewers,
On behalf of my co-authors, I am submitting the enclosed material “Rapid Identification of Constituents in Polygonatum cyrtonema Hua Using UHPLC-Q-Exactive Orbitrap Mass Spectrometry” for possible publication in Molecules.
We have reviewed the revision version of the manuscript and approve it for publication. To the best of our knowledge and belief, this manuscript has not been published nor is it being considered for publication elsewhere.
Your comments were highly insightful and enabled us to greatly improve the quality of the manuscript. In the following pages are our point-by-point responses to each of the comments. The revision has been highlighted in red font in the manuscript.
Comments 1: In this manuscript, Yang and co-workers described the identification of 153 species contained in Plygonatum cyrtonema Hua (PCH) by the combination of chromatographic separations and Mass Spectrometry determination in order to unravel the basis of medical properties of this PCH. Despite the valuable work on species identification based on MS results, I still have some questions that need to be resolved.
- The introduction section explains the main aim of the deeper understanding of the medical properties of PCH through the species identification contained in this herb. However, there is a lack of information on previous efforts made by other researchers to identify the composition of PCH. A brief review of previous experiments that were attempting to determine some of these compounds and the techniques used should be included to better understand the state of the art and the novelty of this work.
Response 1: Thank you for your valuable and thoughtful comments. Research on the chemical constituents of PCH has been added to the introduction section of the article.
Comments 2: Figure 1 shows the six high-resolution extracted ion chromatograms of the six extracts obtained after separation on D101 resin. However, there is no indication of which extract (labelled W, T, S, F, E and N in section 4.2) corresponds to each of the chromatograms shown (A-F).
Response 2: Thank you for your valuable and thoughtful comments. In Figure 1, (A) and (B) represent the Extracted Ion Chromatograms(EIC) of the compounds detected in Extract S under positive and negative ion modes, respectively. (C) and (D) represent the EICs of the compounds detected in Extract F under positive and negative ion modes, respectively. (E) and (F) represent the EICs of the compounds detected in Extract N under positive and negative ion modes, respectively. Extracts S, F, and N collectively contain 153 identified compounds. Extracts W, T, and E are not listed because their compounds overlap with those in Extracts S, F, and N. Additionally, following the suggestion of another reviewer, we have moved Figure 1 to the supplementary materials (Figure 1S).
Comments 3: Furthermore, the description of each chromatogram (A-F) used in the figure caption includes the numbers of the masses detected, but it is somewhat confusing to use this as a label. In addition, there are peaks whose m/z are given in Table 1 but are not mentioned in these labels in Figure 1.
Response 3: Thank you for your valuable and thoughtful comments. We have simplified the figure captions by removing the m/z information. The Arabic numerals in the figure represent the identifiers for the 153 compounds. You can find detailed information, including the m/z or other specifics, in Table 2S by using these numerical identifiers. Additionally, following the suggestion of another reviewer, we have moved Figure 1 to the supplementary materials (Figure 1S).
Comments 4: Also in Figure 1, the axis labels and values are not visible because they are too small. This makes identification difficult.
Response 4: Thank you for your valuable and thoughtful comments. We have increased the size of the numbers in Figure 1 to improve readability and facilitate easier identification of each compound's identifier. Additionally, following the suggestion of another reviewer, we have moved Figure 1 to the supplementary materials (Figure 1S).
Comments 5: The methodological section explained the sample preparation procedure. However, despite the explanation of the two separation techniques used (D101 macroporous resin and reversed phase chromatography), the total time required to carry out the whole chromatographic analyses, 6 extract D101 isolation and sequential UHPLC-Q-Orbitrap MS, was not given.
Response 5: Thank you for your valuable and thoughtful comments. In the Methods, the information on the total run time for the mass spectrometry analysis has been added (30 minutes). Regarding the time for the macroporous resin separation experiment, we have added detailed descriptions in the method and discussion section. These descriptions include the specific time allocations for each step of the experiment, such as approximately 8 hours for the separation experiment, about 8 hours for drying, and the time required for UHPLC-Q-Exactive Orbitrap MS analysis. Through these descriptions, readers can gain a clearer understanding of the overall experimental timeline and its operational simplicity and efficiency.
Round 2
Reviewer 1 Report
Comments and Suggestions for Authors
In topic 4.2 Preparation of Standard and Sample Solutions, it is necessary to insert the proportion used Sample A:resin. What is the mass of Sample A applied to the column? How much resin was used?
With this topic fixed, the article is complete.
Author Response
Comments 1: In topic 4.2 Preparation of Standard and Sample Solutions, it is necessary to insert the proportion used Sample A:resin. What is the mass of Sample A applied to the column? How much resin was used?
Response 1: Thank you for your valuable and thoughtful comments. The mass of Sample A and resin were added in section 4.2 as following sentence “a total of 2 g sample A was separated by chromatography using 200 g D101 macroporous resin”